## RESEARCH ARTICLE

# Uncovering mitotic ultrastructure in the native hair follicle using volume electron microscopy

Nickhil Jadav[1,*], Sailakshmi Velamoor[2,*], Niki Hazelton[3], Karen Reader[4], Duane Harland[5,‡] and Mihnea Bostina[1,3,‡]

## ABSTRACT

The hair follicle undergoes repeated cycles, with anagen representing the active growth phase. During anagen, transit amplifying cells within the germinative matrix at the follicle bulb drive rapid proliferation for hair growth. This region exhibits some of the highest mitotic rates observed in any tissue, offering a rare opportunity to study mitosis in its native epithelial context, previously studied only in cultured cell lines. We applied volume electron microscopy to intact chemically fixed hair follicles enabling exceptional ultrastructural preservation of the entire mini-organ. Morphometric analysis revealed stage-specific changes in chromosomal and organelle volume and spatial distribution, highlighting coordinated roles for the mitochondria, vesicles and endoplasmic reticulum, and enabled, to our knowledge, the first ultrastructure-based karyotype of ovine chromosomes. This work advances understanding of mitosis by resolving ultrastructure in a highly proliferative, spatially constrained epithelial microenvironment, demonstrating the power of serial block face scanning electron microscopy to bridge *in vitro* models and native tissue architecture.

KEY WORDS: Volume electron microscopy, Mitosis, Chromosomes, Cellular ultrastructure, Transit amplifying cells, Hair follicle

## INTRODUCTION

Hair follicles (HFs) are specialised epidermal appendages that cycle between active (anagen) and resting (telogen) phases. During the anagen phase, follicles undergo rapid growth, with epithelial keratinocytes in the follicular bulb actively proliferating to produce hair (Harland, 2018; Plowman and Harland, 2018). These keratinocytes undergo mitosis in the germinative matrix (GM) of the HF, giving rise to a spatially concentrated population of transit amplifying cells (TACs) (Potten, 1981). TACs are an intermediate cell population that plays a vital role in follicular growth by rapidly

[1]Department of Microbiology and Immunology, University of Otago, Dunedin, New Zealand. [2]Department of Biochemistry and Molecular Biology, Biomedicine Discovery Institute, Monash University, Victoria 3168, Australia. [3]Otago Micro and Nano Imaging Electron Microscopy Unit, University of Otago, Dunedin, New Zealand. [4]Department of Pathology, University of Otago, Dunedin, New Zealand. [5]Proteins & Metabolites, Bioeconomy Science Institute, Lincoln 7608, New Zealand.
*These authors contributed equally to this work

‡Authors for correspondence (duane.harland@agresearch.co.nz; mihnea.bostina@otago.ac.nz)

D.H., 0000-0002-1204-054X; M.B., 0000-0003-3621-3772

proliferating before differentiating into mature cells (Harland, 2018; Harland and Plowman, 2018), with the boundary between proliferation and differentiation demarked by 'Auber's critical level' (also known as Auber's line). The TACs within HFs exhibit one of the highest mitotic rates of any organ (Martel et al., 2024), highlighting the crucial role of mitosis in hair growth and regeneration (Fig. S1).

Although mitosis has been widely studied across various cell types, the chromosomal dynamics and organelle interactions during mitosis in intact HFs, particularly within TACs, remain poorly understood. Traditional techniques such as transmission electron microscopy (TEM) have provided valuable insights into mitotic chromatin condensation and the disassembly of cellular structures (Barnicot and Huxley, 1965; Chen et al., 2017; Liang et al., 2015; Mankouri et al., 2013; McEwen et al., 2007; Mora-Bermúdez et al., 2007; Pavelka and Roth, 2010; Sattler et al., 1988), but they are limited by their inability to capture the three-dimensional (3D) ultrastructure of chromosomes and organelles (Jadav et al., 2023; Shemilt et al., 2014; Velamoor et al., 2022). This is largely due to the thin-sectioning nature of TEM, which hinders the observation of complex spatial relationships between cellular components.

Recent advancements in volume electron microscopy (vEM), specifically serial block-face scanning electron microscopy (SBF-SEM), have enabled significant progress in the study of mitosis. In contrast to light-based volume microscopy methods, SBF-SEM offers high-resolution imaging of intact tissue blocks, enabling 3D examination of chromosomal behaviour, organelle distribution and the spatial interactions between cellular components during mitosis (Chen et al., 2017; Sajid et al., 2021; Yusuf et al., 2022). These advancements enhance our understanding of mitotic processes by enabling comprehensive imaging of cellular structures in their natural environment, facilitating the detailed observation of mitotic progression, chromosomal organisation and organelle dynamics (Booth et al., 2016; Ferrandiz and Royle, 2023; Jadav et al., 2023; Sajid et al., 2021; Yusuf et al., 2022).

In this study, we utilised SBF-SEM to investigate the 3D dynamics of mitosis in the TACs of intact HFs. Our goal was to gain a deeper understanding of the spatial relationships between chromosomes and surrounding organelles, with particular focus on the interactions between TACs, chromosomal volume changes and mitochondrial dynamics during mitosis. By examining these processes within the context of native highly proliferative tissue, we aim to offer new insights into the cellular organisation and dynamics of HF TACs during rapid proliferation, contributing to a broader understanding of mitotic processes in living tissues.

## RESULTS

We employed SBF-SEM to investigate the dynamics of cell division in ovine HFs and captured high-resolution images of all five stages of mitosis – prophase, metaphase, anaphase, telophase and

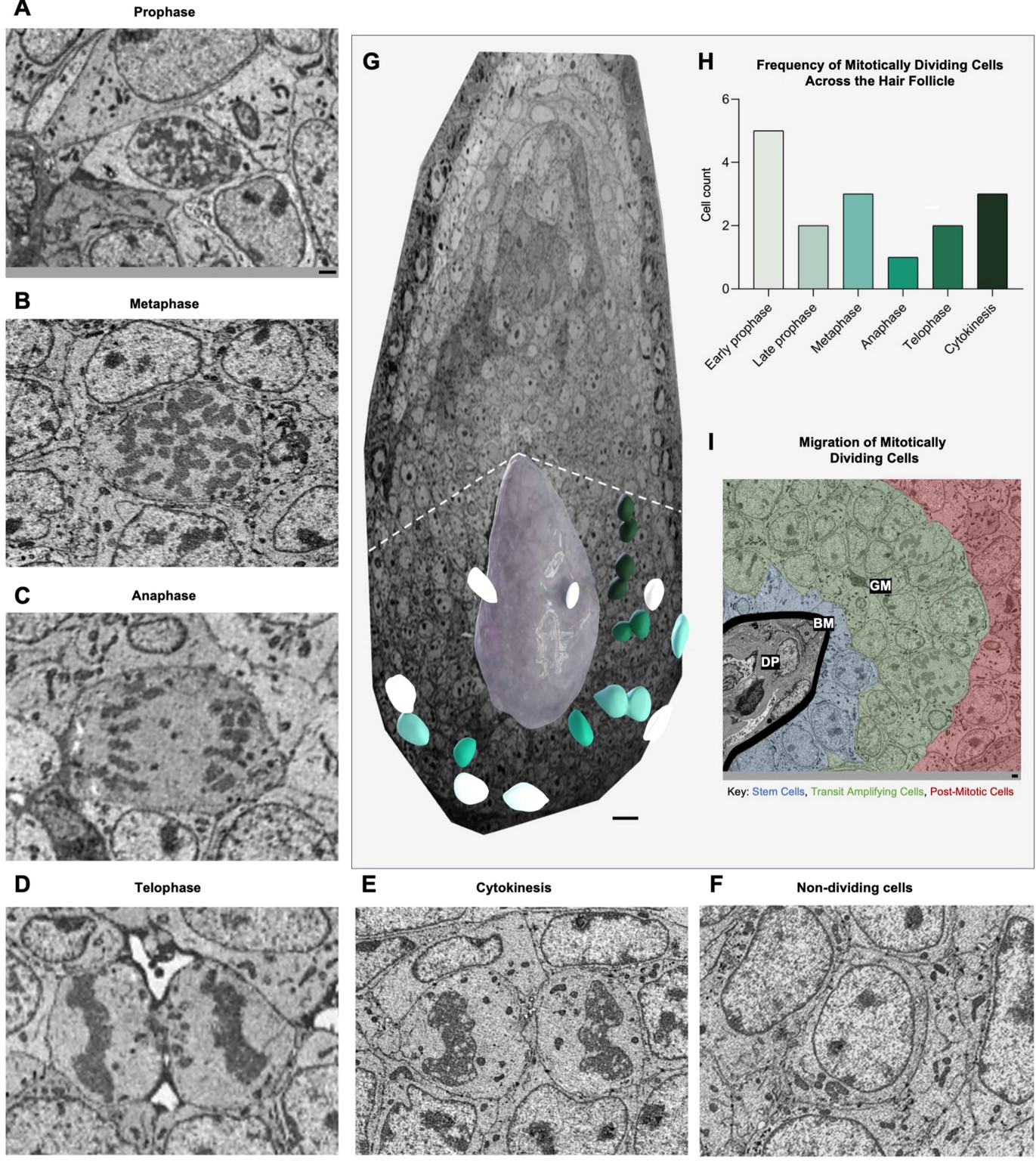

**Fig. 1. Dynamics of cell division and differentiation in HF regions.** SBF-SEM was used to obtain all micrographs. Representative micrographs of cells during different mitotic phases: (A) prophase, (B) metaphase, (C) anaphase, (D) telophase and (E) cytokinesis. (F) Micrograph of non-dividing cells. (G) Mitotically dividing cells and the dermal papilla (DP, grey) superimposed on the z-stack. The green colour gradient indicates progression from prophase to cytokinesis. The white dashed line indicates the theoretical location of Auber's critical level. (H,I) Frequency (H) and migration (I) of mitotically dividing cells within a ovine HF. Data shown are from a single representative HF (one sheep, two separate hair follicles; see Fig. S3). The zones are as follows: DP, basement membrane (BM), and germinative matrix (GM). Cell types: stem cells (blue), transit amplifying cells (green) and post mitotic cells (red). Scale bars: 1 µm (A–F,I); 10 µm (G).

cytokinesis (Fig. 1A–E) – from two separate HFs (Fig. S3). This provided insights into the morphological changes associated with cell division in an intact mini organ.

Mitotically active cells were localised within the germinative matrix (Fig. 1G,H) and exhibited directed centripetal migration as they progressed through different phases (Fig. 1I). We analysed a

total of nine dividing cells and eight non-dividing cells in detail across all mitotic phases (Fig. 1F; Table S2). We reconstructed and rendered each dividing cell in three dimensions to assess the organisation of nuclear, chromosomal and subcellular structures (Fig. 2A,B; Figs S4, S6).

## Volumetric and spatial organisation of chromosomal and nuclear structures

We quantified volumetric changes in cellular, nuclear and chromosomal compartments across different phases of mitosis to assess structural organisation during division (Fig. 2C–I). Dividing cells exhibited significantly larger cellular and nuclear volumes compared to those in non-dividing cells ($P<0.01$). Nuclear volume progressively decreased from prophase I to cytokinesis, whereas chromosomal volume remained relatively stable following a marked increase from prophase I to prophase II. Cellular volume remained elevated during the late stages of prophase (II and III) before decreasing in metaphase and stabilising through to cytokinesis.

We also measured stain intensity to assess chromosomal condensation during mitosis (Fig. 2D). Nuclear stain intensity was highest during interphase, with a significant decrease observed in prophase ($P<0.0001$). Chromosomal stain intensity progressively declined from metaphase to cytokinesis, indicating ongoing chromosomal decondensation as cells completed division.

We also assessed the displacement of chromosomes from the centre of the cell (Fig. 2E). Chromosomal displacement increased steadily from prophase through to cytokinesis, with the largest distances recorded in telophase and cytokinesis. This pattern suggests an outward movement of chromosomes as division progressed. To measure the distance of each chromosome from the centre of the cell (Fig. 2F), we analysed two metaphase cells as they represent the most clearly defined mitotic stage during which chromosome morphology remains relatively constant. Chromosomes displayed a wide range of radial positions, with some positioned centrally and others more peripherally, indicating a non-random spatial arrangement.

We also constructed chromosome distance matrices to visualise spatial relationships between chromosomes (Fig. 2G), focusing on the six largest and six smallest chromosomes. Although overall patterns were similar between the two cells, notable differences in chromosome–chromosome distances were evident, suggesting variability in chromosomal organisation between individual cells. To further examine spatial organisation relative to chromosome size, chromatids were assigned to karyotypic groups based on reconstructed volume, allowing visualisation of their positional relationships within the dividing cell. These findings were further supported by network maps, which revealed that in both cells, the X and Y chromosomes maintained multiple connections, indicating that the sex chromosomes actively participated in spatial interactions within the metaphase network. Specifically, the X chromosome, positioned predominantly near the cell periphery, clustered mainly with large autosomes, whereas the Y chromosome associated more closely with smaller autosomes. High-resolution 3D reconstructions confirmed the positioning of the sex chromosomes relative to large and small autosomes (Fig. 2I). Consistent with these findings, volume-assigned karyotypic reconstructions demonstrated organised chromatid positioning in metaphase and somewhat equal segregation of chromosome groups during anaphase (Fig. S5).

## Karyotypic reconstruction of sheep chromosomes during metaphase

To characterise the karyotype of sheep HF cells, individual segmented chromosomes from SBF-SEM were analysed (Fig. 3A; Fig. S8). Chromosome identification was achieved by integrating volumetric analysis, spatial orientation and distinct morphological features (Fig. 3A,B). Chromosomes from metaphase were arranged in descending order of volume and surface area to facilitate karyotyping. Based on centromere position, chromosomes were categorised into metacentric, submetacentric and acrocentric groups, which served as secondary identification markers. Using these criteria, chromosomes were classified into four distinct groups and summarised in a 3D karyotype (Fig. 3A).

Of the 26 autosomes, the three largest chromosomes (1–3) were metacentric. The remaining 23 pairs were acrocentric and were further subdivided based on volume. Acrocentric group I included six pairs (chromosomes 4–9) with volumes greater than 3 µm³. Acrocentric group II comprised eight pairs (chromosomes 10–17) with volumes between 2 and 3 µm³, whereas acrocentric group III contained nine pairs (chromosomes 18–26) with volumes less than 2 µm³. The X chromosome was identified as the largest submetacentric chromosome, and the Y chromosome as the smallest acrocentric chromosome.

We next compared the volume, surface area and surface area-to-volume (SA/V) ratio across karyotypic groups (Fig. 3C–E). The three metacentric autosomes exhibited the largest individual volumes and surface areas, with the lowest SA/V ratios, indicating a compact and condensed morphology. Chromosomes in acrocentric group I also displayed large volumes (>3 µm³) and high surface areas but had moderately higher SA/V ratios compared to the metacentric group, suggesting a slightly less compact structure.

Chromosomes in acrocentric group II showed intermediate volumes (2–3 µm³) with reduced surface areas and further increased SA/V ratios, consistent with a more elongated morphology. Acrocentric group III displayed the smallest volumes (<2 µm³) and surface areas, but the highest SA/V ratios among autosomes, reflecting a relatively greater surface complexity.

The X chromosome exhibited a large volume and surface area comparable to acrocentric group I chromosomes, with a similar SA/V ratio. In contrast, the Y chromosome had the smallest volume and surface area, but a high SA/V ratio relative to its size, consistent with a less condensed structure.

We next assessed the relative stain density across karyotypic groups (Fig. 3F). Stain intensity, normalised to total chromosomal stain per metaphase cell, was broadly proportional to chromosome volume. Larger metacentric chromosomes and chromosomes in acrocentric group I displayed higher integrated stain intensities compared to smaller acrocentric chromosomes and the Y chromosome. However, several small acrocentric chromosomes exhibited relatively high stain densities compared to their volumes, suggesting local chromatin compaction or differences in chromatin structure. The X chromosome displayed a stain intensity comparable to larger acrocentric chromosomes, whereas the Y chromosome showed the lowest relative stain intensity.

## Organelle dynamics during mitosis in sheep HF cells

We conducted a comprehensive analysis of mitochondria, endoplasmic reticulum (ER) and vesicles across distinct mitotic phases in sheep HF cells using high-resolution 3D reconstruction (Fig. 4A,B). Organelle morphology and distribution were examined in cells at prophase (I), metaphase (II), anaphase (III), telophase (IV), and cytokinesis (V). Subcellular analysis revealed both volumetric and numerical shifts during mitotic progression (Fig. 4C,D).

Combined mitochondrial volume gradually increased from early prophase (I) and nearly doubled by late prophase (III), a trend maintained through metaphase. Although there were no significant volumetric differences between metaphase and later stages (anaphase

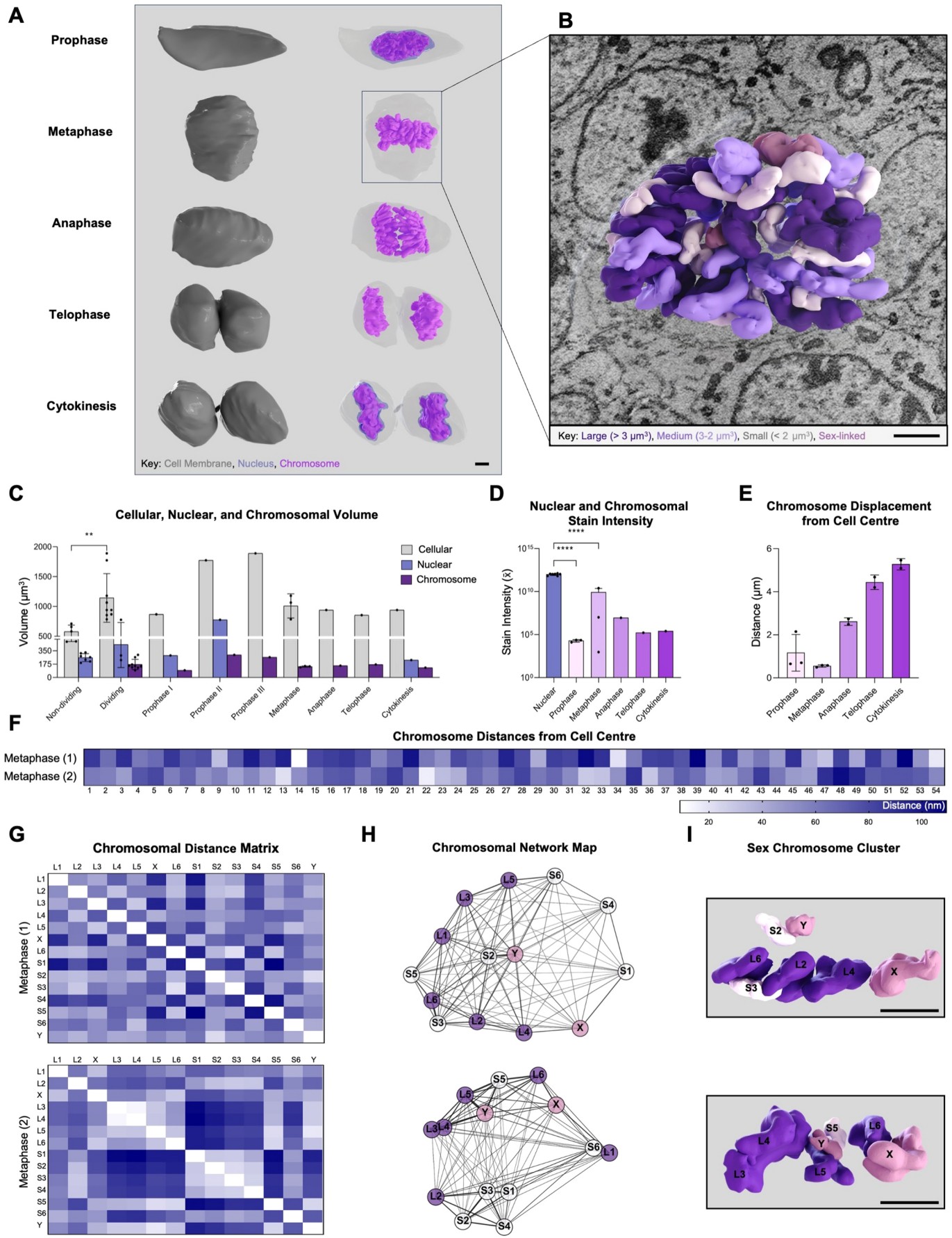

**Fig. 2.** See next page for legend.

**Fig. 2. Mitotic cellular components analysis in HF cells.** (A) 3D reconstruction of a mitotic cells from prophase to cytokinesis, highlighting the cell membrane (grey), nucleus (blue) and chromosomes (purple). (B) Metaphase cell superimposed on a z-stack slice showing chromosomal volumetric progression; descending shades of purple denote size (large, >3 µm$^3$; medium, 2–3 µm$^3$; small: <2 µm$^3$), with sex-linked chromosomes in dark pink. (C) Volumetric analysis (µm$^3$) of cellular, nuclear and chromosomal compartments of non-dividing and dividing cells. (D) Mean stain intensity of nuclear and chromosomal structures. (E) Chromosomal displacement from cell centre (µm). (F–I) Chromosomal spatial analysis in metaphase cells (n=2). (F) Heatmap of chromosome distances (nm) from the cell centre, with chromosomes numbered 1–54 by descending volumes. (G,H) Chromosome distance matrix (nm) (G) and network map (H) of the six largest (L1–L6), smallest (S1–S6), and sex-linked chromosomes (X and Y) ranked by descending volume. (I) Reconstructed chromosome cluster of sex-linked and associated chromosomes from GMM clustering algorithm. Scale bars: 2 µm. Data presented as means±s.d. Individual data points are shown; sample sizes (n) are detailed in Table S2. **P≤0.01, ****P≤0.0001 (two-tailed unpaired parametric t-test with Welch's correction).

and cytokinesis), mitochondrial counts followed a similar trajectory – rising steadily from prophase, peaking at metaphase and remaining relatively stable through cytokinesis (Fig. 4D). Spatially, mitochondria exhibited dynamic reorganization. The K-mean-defined mitochondrial cluster count per cell significantly increased during anaphase (Fig. 4E) and telophase compared to that in earlier phases and non-dividing cells (Fig. 4G; Fig. S9A). Although mitochondria appeared more compact during metaphase, this difference in clustering density was not statistically significant (Fig. 4F). Mitochondrial hotspot mapping using 3×3 heatmap analysis (Fig. 4H) revealed that the highest mitochondrial density occurred at the cell centre during telophase. This central concentration was further illustrated in reconstructed telophase cells, where a red-to-yellow gradient depicted localized mitochondrial enrichment (Fig. 4I). These patterns suggest mitochondria are strategically repositioned during mitosis, likely supporting localized energy demands and structural remodelling.

ER volume increased modestly from prophase to metaphase, followed by a significant rise during anaphase that was sustained through to cytokinesis (Fig. 4C). However, the number of ER fragments showed an inverse trend – gradually decreasing from prophase to metaphase, then sharply increasing during anaphase (Fig. 4D). This pattern is consistent with ER fragmentation and redistribution, aligning with its known role in mitotic spindle association and intracellular compartmentalization.

Vesicular volume and count peaked during metaphase, followed by a reduction in telophase and cytokinesis (Fig. 4C,D). Unlike mitochondria, vesicles displayed relatively stable numbers throughout mitosis, with the most notable organizational change occurring in the final mitotic stages. Spatial clustering analysis revealed that vesicle clusters were loosely dispersed in early mitosis, becoming more compact and localized during telophase and cytokinesis (Fig. S9B). Inter-vesicle distances were shortest in anaphase, indicating directed trafficking or cortical accumulation during spindle reorganization. Vesicle hotspot maps highlighted peri-spindle and cortical zones of high density during late mitosis – patterns absent in non-dividing cells (Fig. S10). These findings suggest that vesicles are redistributed for membrane remodelling and abscission, reflecting their late-stage functional role. Taken together, all these data reveal a highly coordinated yet organelle-specific remodelling program during mitosis.

## DISCUSSION

Traditionally, ultrastructural studies of mitosis have relied heavily on electron microscopy in cultured cell lines (Ahmed, 1940; Chen et al., 2017; Liang et al., 2015; Sajid et al., 2021; Yusuf et al., 2022), with fewer investigations focusing on tissue sections (Adolph, 1989; Barnicot and Huxley, 1965). Although TEM, including electron tomography, offers valuable ultrastructural insights, its limitations in sectioning thickness, orientation of sections with respect to microanatomy and the high level of resourcing required to carry generate volume data (e.g. serial tomograms) challenge comprehensive analysis at sample sizes needed for statistically robust experiments. Recent advances in 3D imaging, particularly automated approaches, such as SBF-SEM, have overcome many of these challenges (Booth et al., 2016; Boucrot and Kirchhausen, 2007, 2008; Chen et al., 2017; Cisneros-Soberanis et al., 2024; Sajid et al., 2021; Yusuf et al., 2022). Here, we present the first comprehensive application of SBF-SEM to analyse the spatial dynamics of mitosis within ovine HF cells.

In the HF, epithelial stem cells reside within the bulge and germinative matrix, where they rapidly proliferate within a confined region of the lower follicle. These stem cells give rise to TACs, which migrate centripetally and upwards towards Auber's line and beyond, where they begin to differentiate into specialised cell types that contribute to the hair shaft and supporting sheath layers (Auber, 1952; Harland and Plowman, 2018; Tang et al., 2024; Yang et al., 2017; Zhang and Hsu, 2017). A similar process occurs in the cornea, where limbal epithelial stem cells generate TACs that proliferate, migrate centripetally and differentiate to replenish the shed epithelium (Cancedda and Mastrogiacomo, 2023; Yoon et al., 2014). This shared stem cell–progenitor structure likely represents a conserved mechanism used by various tissues to maintain epithelial integrity. In our study, we observed that cells migrate outwards and upwards from the niche as mitosis progresses from prophase to cytokinesis, suggesting a well-coordinated spatiotemporal pattern of division. This behaviour aligns with established models of hair growth during anagen (Tang et al., 2024; Zhang and Hsu, 2017) and suggests that mitosis is intrinsically linked to directed cell migration.

High-resolution 3D reconstructions allowed us to comprehensively characterise the cellular, nuclear, chromosomal and organelle dynamics across all mitotic stages. We observed substantial remodelling of nuclear and subcellular structures (Fig. S4), demonstrating that mitotic progression in HF cells involves highly coordinated morphological and organisational changes. Segmentation and 3D rendering demonstrated that nuclear volume nearly doubled during prophase and decreased markedly by telophase, consistent with chromatin condensation and nuclear envelope breakdown (Sajid et al., 2021). Chromosomal volume, in contrast, remained relatively stable following an early prophase increase, suggesting that major condensation processes are completed before metaphase (Booth et al., 2016; Cisneros-Soberanis et al., 2024). In addition to corroborating with earlier observations in cultured cells (Booth et al., 2016; Boucrot and Kirchhausen, 2007, 2008; Chen et al., 2017; Sajid et al., 2021; Yusuf et al., 2022), these data provide direct in situ evidence that nuclear and chromosomal changes follow tightly regulated patterns. Although our analysis is based on spatial snapshots rather than live temporal imaging, these reconstructions reveal how subcellular organisation aligns with tissue-level behaviours, including oriented division and directional displacement. Together, these observations indicate that mitotic structures and organisation are conserved in situ and closely integrated with the proliferative epithelial niche.

Spatial analyses showed that chromosomes gradually displaced outward from the cell centre as mitosis progressed, with maximal displacement observed in telophase and cytokinesis. Radial mapping during metaphase revealed a non-random distribution, with larger chromosomes located more peripherally and smaller

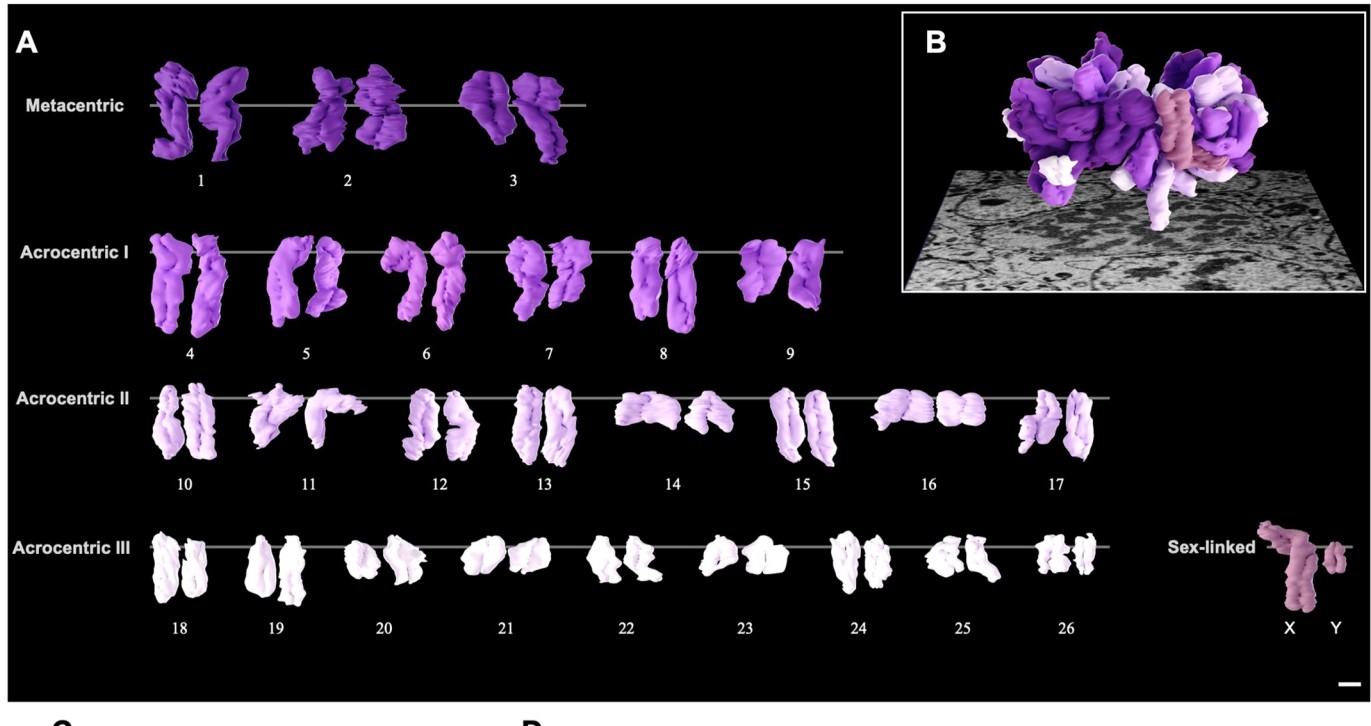

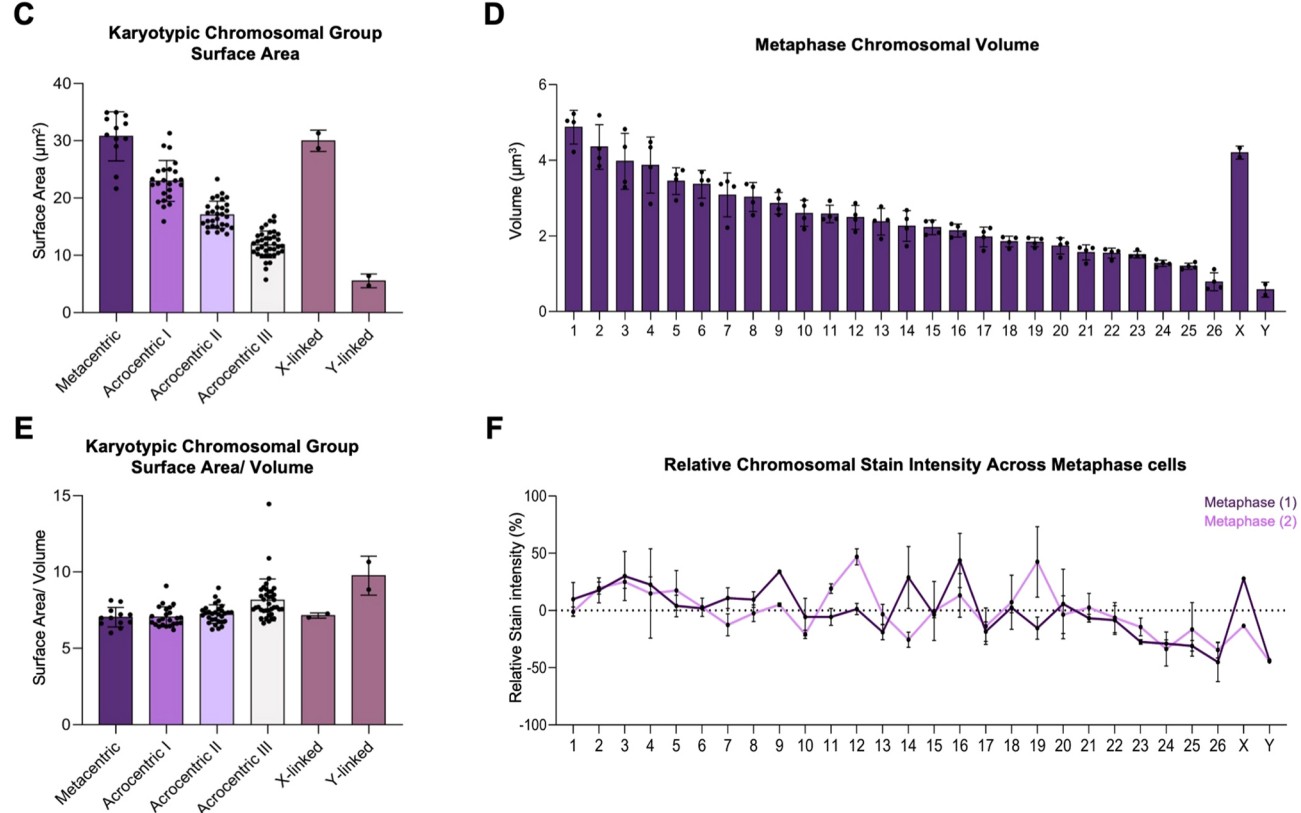

**Fig. 3. Karyotypic mapping of ovine chromosomes based on volume.** (A) 3D reconstruction of diploid chromosomes during metaphase. Karyotype shown grouped by volume and centromere position: metacentric (purple), acrocentric grouped I–III by descending volume (lilac to white), and sex-linked chromosomes (dark pink) presented as X and Y. (B) Metaphase reconstruction of chromosomes coloured via its karyotypic identity superimposed on a micrograph. Chromosomes were matched based on volume, centromere position, spatial orientation and distinct morphology features. Scale bar: 1 µm. (C–F) Chromosomal analysis of metaphase cells (*n*=2). (C) Surface area (µm$^2$) comparison between karyotypic groups; (D) volume (µm$^3$) analysis of diploid chromosomes; (E) surface area-to-volume ratio between karyotypic groups; and (F) normalised stain intensity analysis of diploid chromosomes, calculated as the average integrated density for each chromosome across the *z*-stack, normalised to the total chromosomal stain intensity. Data are presented as means±s.d. Individual data points are shown. See Table S3 statistical details for C and E.

Journal of Cell Science

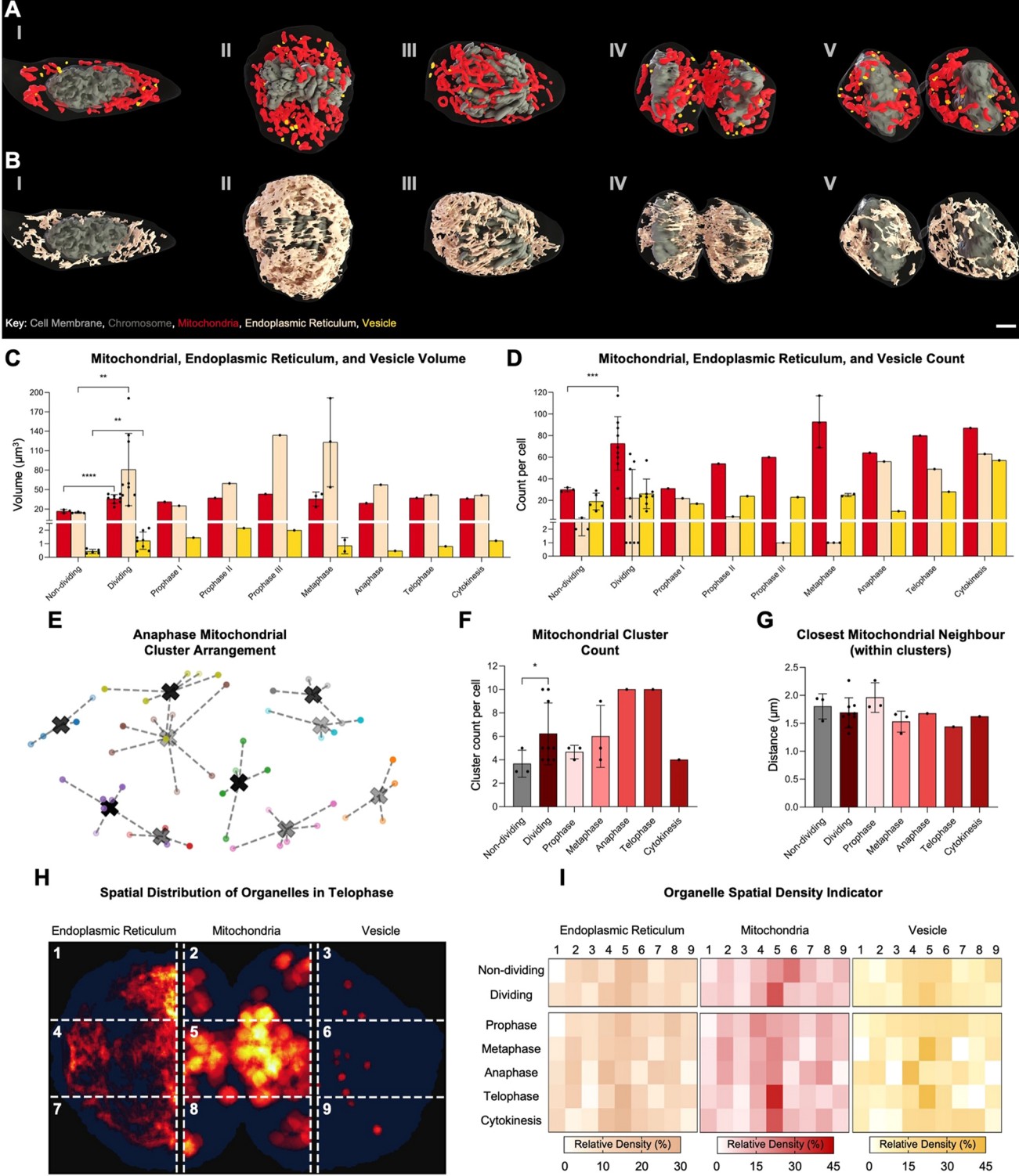

**Fig. 4. Mitochondrial, ER and vesicle analysis of HF cells.** (A,B) Reconstructed sub-cellular organelles at different mitotic phases: (I) prophase, (II) metaphase, (III) anaphase, (IV) telophase and (V) cytokinesis. Cell membrane (dark grey), chromosomes (olive grey), mitochondria (red), vesicles (yellow), endoplasmic reticulum (ER, cream). (C,D) Subcellular analysis of mitochondria (red), ER (cream) and vesicles (yellow), showing (C) volume and (D) count. (E–G) Mitochondrial cluster analysis, extracting the individual mitochondrial centre of mass (COM) and applying silhouette and k-means algorithm. (G) 3D cluster arrangement example within an anaphase cell. Different colours denote different clusters, with the cluster centre marked by an 'X'. (F) Cluster count per cell. (G) Closest mitochondrial neighbour distance (µm) within clusters. (H,I) Spatial organelle density analysis. (H) Organelles (mitochondria, ER, vesicles) are split into 3×3 regional heatmaps displayed in vertical columns per organelle. Heatmaps show z-corrected organelle density across aligned cells (the x-axis is the longest axis, then the y-axis). (I) Organelle-specific hotspot maps for mitochondria (red), ER (cream) and vesicles (yellow), highlighting areas of high spatial density overlaid on a telophase cell membrane (dark blue). White indicates low density, and progression to the organelle-specific colour denotes increasing density. Scale bar: 2 µm. Data presented as means±s.d. Individual data points are shown; sample sizes (n) are detailed in Table S2. *P≤0.05; **P≤0.01; ***P≤0.001; ****P≤0.0001 (two-tailed unpaired parametric t-test with Welch's correction). See Fig. S9 and S10 for further examples of results from E and H.

chromosomes concentrated near the centre. This pattern matches previous observations suggesting that gene-rich chromosomes localise towards the nuclear interior and gene-poor chromosomes towards the periphery (Chen et al., 2017; Cisneros-Soberanis et al., 2024; Sajid et al., 2021). Notably, we found that the X and Y chromosomes maintained multiple spatial interactions with both large and small autosomes. In contrast to previous findings (Sajid et al., 2021), we observed the Y chromosome positioned predominantly near the centre of the cell. This might indicate tissue-specific differences in chromosome organisation during mitosis or species-specific nuclear architecture in ovine HF cells. Further research is needed to understand the biological importance of the sex chromosome positioning.

Using volumetric and morphological parameters, we successfully identified all 54 ovine chromosomes in two metaphase cells. SBF-SEM has been shown to resolve chromosomal structure and spatial organisation (Booth et al., 2016; Booth and Earnshaw, 2017; Cisneros-Soberanis et al., 2024; Sajid et al., 2021; Yusuf et al., 2022), and here we demonstrate for the first time its application for high-resolution karyotyping *in situ* within an intact mini-organ, preserving native tissue architecture and providing a more biologically relevant context than conventional culture-based approaches. By resolving chromosomal size, centromere position, and structural class, this approach enables comprehensive analysis of nuclear organisation within native tissue and provides a platform to investigate how chromosome architecture correlates with mitotic dynamics and tissue-level behaviour.

Organelle analysis revealed highly coordinated remodelling to meet the energetic and spatial demands of cell division. Spatial organisation of mitochondria was in agreement with earlier studies (Carlton et al., 2020; Lawrence and Mandato, 2013; Martínez-Diez et al., 2006). That is, mitochondrial volume and cluster formation peaked during metaphase and telophase, reflecting an increased demand for ATP and localised energy production during spindle formation and cytokinesis (Carlton et al., 2020; Zhao et al., 2024). Additionally, mitochondrial recruitment near the cleavage furrow might help ensure equitable mitochondrial inheritance (Christiansen, 1949; Lawrence and Mandato, 2013).

Remodelling of the ER during mitosis was characterized by a modest increase in ER volume from prophase to metaphase, followed by a rapid expansion during anaphase. Meanwhile, the number of ER fragments decreased early on but rose significantly at later stages. This pattern reflected nuclear envelope breakdown in early mitosis, where inner and outer nuclear membranes integrated into the ER, expanding the network (Carlton et al., 2020; Champion et al., 2017; Zhao et al., 2024). During late anaphase and telophase, ER membranes reorganised to enclose daughter chromosomes and re-form the nuclear envelope (Carlton et al., 2020; Champion et al., 2017). Together, these findings highlight that the ER has a central role in maintaining nuclear integrity and facilitating mitotic progression.

Vesicular dynamics in our study indicated active membrane trafficking and remodelling, particularly in preparation for cytokinesis during late mitosis. The Golgi complex is known to break down into small vesicles during prometaphase, undergoing further fragmentation in metaphase. These vesicles dispersed throughout the cytosol (Persico et al., 2009) before reforming during telophase (Carlton et al., 2020; Pavelka and Roth, 2010; Thyberg and Moskalewski, 1998). The marked increase in vesicle number during metaphase observed in our study aligns with this fragmentation process. Collectively, these findings provide direct *in situ* evidence that organelle remodelling during mitosis is highly coordinated,

spatially organised, and closely aligned with cellular energetic and structural demands. Extending observations from cultured cells to native tissue, they demonstrate that subcellular dynamics are integrated with tissue-specific architecture and cellular context, highlighting the universal importance of organelle coordination for accurate mitotic progression and efficient post-mitotic recovery.

In conclusion, our study showcases the power of SBF-SEM for high-resolution, 3D mapping of mitotic progression in ovine HF cells. Notably, this is the first *in situ* ultrastructural analysis of cell division stages captured within a complex, intact mini-organ – rather than in conventional 2D or 3D culture systems. Although previous studies have imaged mitosis in cultured cells, these models fail to fully recapitulate the spatial and cellular complexity of native tissue architecture. In contrast, our approach enables direct observation of mitotic progression within the germinative matrix, a spatially restricted zone crucial for hair fibre formation.

The novelty of our dataset further highlights our ability to reconstruct complete karyotypes from dividing cells *in situ*. This unprecedented level of ultrastructural detail provides valuable insight into nuclear architecture, chromatin dynamics and organelle organisation during hair fibre development. These findings not only advance our understanding of mitosis within a self-renewing epithelial mini-organ but also have broader implications for developmental biology, keratin self-assembly and fibre formation – with potential applications in both the agricultural and biomedical sectors. In future, integrating SBF-SEM with complementary live-cell imaging or molecular techniques could further enhance understanding of the temporal dynamics underlying mitotic progression.

## MATERIALS AND METHODS

### Wool follicle collection and processing

The follicles were isolated from 4 mm punch (Kai Industries) biopsies taken from cleaned skin (shaved and 70% ethanol sterile wiped) under a neural block created using local anaesthetic (2% lignocaine, Phoenix Pharm NoPaine 2%) from the mid-sides of two composite sheep (*Ovis aries*) as described in Velamoor et al. (2020) at AgResearch's research farms at Lincoln, New Zealand, 21 April 2021 (NZ autumn). Collection procedures complied fully with New Zealand law (Permit 14810 using standard operating procedure SHEEP 14, AgResearch Invermay Animal Ethics Committee).

All fixation and processing steps were carried out using EM grade chemicals. On collection, biopsies were immediately placed into excess (4 ml) freshly made fixative [Karnovsky's: 2.5% glutaraldehyde (Electron Microscopy Sciences, L0001340); 2% formaldehyde (Proscitech, C006); 0.05 M sodium cacodylate (Sigma, CO250) buffer, three drops 30% $H_2O_2$ (Fisher, H/1820/15) added immediately before use], which was also gently injected into the biopsy using a fine-gauge insulin needle.

Follicles were isolated from skin biopsies under more fixative using fine tweezers, needles and scalpels in Petri dishes under a dissecting microscope at ambient temperature (~20°C), which was completed in 5 h. Isolated follicles were returned to fresh fixative and incubated for 17 h at 4°C before being washed three times for 15 min in chilled (4°C) 0.1 M sodium cacodylate buffer with 7.5% sucrose (Sigma, S9378) and stored for several weeks at 4°C in the same buffer.

SBF-SEM followed a previously established protocol (Hua et al., 2015; Mikula and Denk, 2015; Polilov et al., 2021; Seligman et al., 1966). All steps were performed on a rotator except for steps involving a water bath or chilling. Follicles were fixed in 2% osmium tetroxide (Electron Microscopy Sciences, 19130) in 0.1 M cacodylate (pH 7.4) in the dark for 90 min. They were then stained with 2.5% ferrocyanide (BDH, 2693300) in 0.1 M cacodylate in the dark for 90 min before being washed twice for 30 min in double distilled water (ddH$_2$O). Samples were then incubated in a saturated thiocarbohydrazide (Sigma-Aldrich, 223220) aqueous solution at 40°C for 45 min and then washed twice for 30 min in ddH$_2$O. They were again incubated in 2% osmium tetroxide in ddH$_2$O in the dark for 90 min, washed twice for 30 min in ddH2O and then transferred into 1% uranyl acetate

(BDH, 10288) in ddH$_2$O at 4°C overnight. The follicles were warmed in the uranyl acetate to 50°C for 120 min, washed twice for 30 min in ddH$_2$O and transferred into prewarmed lead aspartate [Sigma, Lead(II) nitrate, 228621; Neo-Froxx GmbH, L-aspartic acid 1205RG100] solution at 50°C for 120 min. The follicles were washed twice for 30 min in ddH2O and then dehydrated in a graded ethanol (Fisher, E/0650DF/17) series (30 min each in 50, 75, 100% ethanol) at 4°C before being dehydrated in 100% acetone (Fisher, A/0600/17) for 60 min. The follicles were then infiltrated in a 1:1 ratio of acetone in EMbed 812 epoxy resin [Electron Microscopy Sciences (EMS), Hatfield, Pennsylvania, USA; product 14120] overnight, and then in pure resin for 6 h before being transferred to embedding moulds and cured at 60°C for 48 h.

Blocks were hand-trimmed to orient each follicle for longitudinal sectioning, and 60-nm and 100-nm sections were cut using an Ultracut UC7 ultramicrotome (Leica, Germany) equipped with a 45° diamond knife (Diatome, Switzerland). Sections were collected onto 100-mesh formvar-coated copper grids and were used primarily to assess the quality of staining. Staining quality was evaluated using a 100-kV transmission electron microscope (Morgagni 268D, Thermo Fisher Scientific, USA), and micrographs were acquired with a CCD digital camera (Tengra, Emsis, Germany) mounted below the phosphor screen. Following this assessment, the blocks were further trimmed for SBF-SEM (Fig. S11).

### Microscopy

Embedded samples were trimmed to an approximately 0.4 mm$^3$ block. CircuitWorks Conductive Epoxy adhesive (Chemtronics; CW2400; Kennesaw, GA, USA) was used according to manufacturing instructions to mount the trimmed block onto the micro module sample holder (Gatan Inc., Pleasanton, California, USA; PEP6590). Once the adhesive had set (~48 h), the resin blocks were coated with a layer of colloidal silver (ProSciTech; EMS12630) to prevent charging. The sample was mounted in the SEM chamber. Images were collected using a Zeiss 300 Serial Block Face SEM with Gatan 3View® system software (Gatan). Electron beam energy of 1.5 keV was applied along with focal charge compensation (44.1% nitrogen; Deerinck et al., 2018) and the block surface was imaged using an OnPoint backscatter detector (Gatan Inc.). Using the in-built ultramicrotome (3View) serial data collection proceeded through automatic cycles of imaging, raising and cutting away of block surface. All subsequent cellular data came from two wool follicles. Across all follicles, a total of 1500 images with a voxel resolution of 30×30×30 nm was collected over a period of 15 h (Table S1). DigitalMicrograph® 3.5 software package was used to acquire the data set (Gatan Inc.).

### Image processing

The acquired images were aligned used the Etomo function within the IMOD 4.11 software suite (Kremer et al., 1996), and Microscopy Image Browser (MIB; MATLAB 2.84; Belevich et al., 2016). To segment the coarse features, the files were cropped and subsequently binned by a factor of three. The built-in Z-factor correction from the IMOD model header dialogue was incorporated into our dataset. The individual cells and subcellular structures, such as the nucleus, chromosomes and vesicles, were then segmented using the Drawing and Interpolator tools in IMOD, maximising accuracy and efficiency. Quantification of the segmented objects was performed using the Imodinfo function within IMOD, providing detailed morphological data, including shape, volume, size, distribution and radial spatial relationships.

Abundant subcellular structures such as mitochondria and ER were segmented using the Trainable Weka Segmentation v4.0.0 plugin available in Fiji 2.16.0 (Arganda-Carreras et al., 2017; Schindelin et al., 2012). After segmentation, reconstructed 3D structures presented in the main figures were then exported from Chimera 1.17.1 (Pettersen et al., 2004) into Blender ver. 5.1 and edited for colour, material, environment and lighting. A schematic summarising the entire workflow from biopsy collection to segmentation and quantification is provided in Fig. S2.

### Data and statistical analysis

The analysis included a variable number of cells, as well as their cellular and subcellular structures. This variability was attributable to two main factors:

(1) the incomplete volume capture of certain structures within the individual cell, and (2) the labour-intensive nature of the manual segmentation process. As a result, not all structures could be analysed in every cell. An example of labelled cellular and subcellular structures is shown in Fig. S7. Details on the number of cells and structures included in the analysis are provided in Table S2.

In instances where duplicated chromosomes were challenging to discern as distinct pairs, several criteria were used to guide the pairing process, prioritising consistency across volume, morphology and spatial proximity. The primary criterion was volume similarity, followed by morphological features, including centromere position and overall chromosome shape. Proximity served as a supporting criterion, where chromosomes located in proximity were more likely to be paired. In cases where all three criteria were aligned, pairing confidence was highest. Where ambiguity remained, reference to published karyotype data from sheep chromosomes (Hansen, 1973) using a light microscope was used to guide expected pairings, including the number and type of chromosomes (e.g. metacentric or submetacentric) and identification of the X chromosome based on morphology and size. Once pairs were established, 3D karyotypes were constructed, and chromosomes were organised within centromere classes in descending order of volume.

Stain intensity within segmented objects was calculated as the cumulative measure of the grayscale pixel values (integrated density) using Fiji's built-in macro functionality. The initial step in this process involved consistent thresholding of the z-stacks derived from all analysed samples. A custom macro (https://github.com/Nickhil-Jadav/vEM-PyTools) was used to isolate the segmented region using math operations, and the integrated density was calculated for each individual slice within the z-stack. These per-slice values were then summed to obtain the total intensity for the entire volume. Batch processing was then applied to run this macro across multiple files, ensuring consistent analysis across all samples.

The centre of mass (COM) of an object was calculated using 3dmod (command line, imodinfo -c), and the relationship between two different objects in 3D space was calculated using a straight-line distance formula:

$$\text{Distance} = \sqrt{(x2 - x1)^2 + (y2 - y1)^2 + (z2 - z1)^2},$$

where $x1$, $y1$ and $z1$ are the coordinates of the first point (COM1), and $x2$, $y2$ and $z2$ are the coordinates of the second point (COM2).

From this analysis, we created a chromosomal distance matrix depicted as a heatmap chromosomal network graph and cluster analysis. Additionally, for the Weka-derived structures data, the Fiji 3D suite plugin (Ollion et al., 2013) was utilised to gather the centroid information. K-means clustering was calculated using a custom Python script (https://github.com/Nickhil-Jadav/vEM-PyTools) with the Scikit-learn package (Pedregosa et al., 2011) and the optimal K value was determined using the silhouette method.

Organelle densities were mapped across cells using a standardised 3×3 matrix applied to z-projected (2D) images of each segmented cell, aligned along the major cellular axis. This projection approach allowed for consistent spatial comparison across cells of varying shapes and mitotic stages. Voxel-corrected densities, adjusted for varying voxel sizes due to differences in binning across stacks, were calculated using a custom Python script (https://github.com/Nickhil-Jadav/vEM-PyTools). The script quantified organelle density within each region of the matrix, enabling an overview of spatial distribution patterns.

Data analysis and visualisation was performed with GraphPad Prism v.8 software. Data was analysed by two-tailed unpaired parametric t-test, supplemented with Welch's correction to account for the potential inequality of variances between the two sample groups. Custom Python tools used in this study, along with additional utilities (see https://github.com/Nickhil-Jadav/vEM-PyTools).

### Acknowledgements
Sample processing was undertaken by Sharon Lequeux, and biopsy collection and follicle isolation by David Scobie and Marina Richena. The authors acknowledge the staff at Otago Micro and Nanoscale Imaging (OMNI) electron microscopy suite for their technical support.

## Competing interests

The authors declare no competing or financial interests.

## Author contributions

Conceptualization: M.B., N.J., S.V., D.H.; Data curation: M.B., N.J., S.V., N.H., D.H.; Formal analysis: N.J., S.V.; Funding acquisition: M.B., D.H.; Investigation: M.B., N.J., S.V., D.H.; Methodology: M.B., N.J., S.V., N.H., D.H.; Project administration: M.B., D.H.; Resources: M.B., D.H.; Supervision: M.B., D.H.; Validation: M.B., N.J., S.V., K.R., D.H.; Visualization: N.J.; Writing – original draft: N.J., S.V.; Writing – review & editing: M.B., N.J., S.V., K.R., D.H.

## Funding

The study was in parts supported by AgResearch through its Smart and Sustainable Biomaterials Programme [New Zealand Government Ministry of Business Innovation & Employment (MBIE), Science strategic investment fund] and by the Wool Research Organisation of New Zealand (WRONZ) and the MBIE via the 'New Materials from Wool' Partnership. Open Access funding provided by University of Otago. Deposited in PMC for immediate release.

## Data and resource availability

All custom macro and scripts used in this study are available at https://github.com/Nickhil-Jadav/vEM-PyTools. All other relevant data and details of resources can be found within the article and its supplementary information.

## First Person

This article has an associated First Person interview with the first author of the paper.

## Peer review history

The peer review history is available online at https://journals.biologists.com/jcs/lookup/doi/10.1242/jcs.264198.reviewer-comments.pdf

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
