## [Peer Review File · Journal of Cell Science]

Uncovering mitotic ultrastructure in the native hair follicle using volume electron microscopy

Nickhil Jadav, Sailakshmi Velamoor, Niki Hazelton, Karen Reader, Duane Harland and Mihnea Bostina

DOI: 10.1242/jcs.264198

Editor: Robert Parton

Review timeline

Original submission:	8 June 2025
Editorial decision:	16 July 2025
First revision received:	8 October 2025
Editorial decision:	24 October 2025
Second revision received:	19 January 2026
Accepted:	19 January 2026

Original submission

First decision letter

MS ID#: jcs.264198

MS TITLE: Decoding hair fibre genesis: 3D mitotic cycle exploration through volume electron microscopy

AUTHORS: Mihnea Bostina; Nickhil Jadav; Sailakshmi Velamoor; Niki Hazelton; Karen Reader; Duane Harland

ARTICLE TYPE: Research Article

Dear Dr Bostina,

We have now reached a decision on the above manuscript.

To see the reviewers' reports and a copy of this decision letter, please go to:

As you will see, the reviewers raise a number of substantial criticisms that prevent me from accepting the paper at this stage. In particular, all three reviewers were concerned about the limited number of samples analysed (and therefore no indication of inter-sample variability). This would need to be addressed in a revised manuscript. As well as addressing the technical concerns of the reviewers it would also be important to outline the general cell biological implications of the findings.

If you think that you can deal satisfactorily with the criticisms on revision, I would be pleased to see a revised manuscript. We would then return it to the reviewers.

Reviewer 1

SUMMARY OF THE ADVANCE MADE IN THIS PAPER AND ITS POTENTIAL SIGNIFICANCE TO THE FIELD

The manuscript by Jadav et al is a gigantic task of reconstructing hair cells in different division steps in 3D-EM. The task is not trivial and the authors did a great job putting together and

analysing the data.

However, I did not see any major contribution to the field of cell biology. It is an amazing work in VolumeEM field, but I believe the impact beyond this is very limited. The number of samples analysed is very small, limiting any interpretation that can be made.

Perhaps the title should be changed. The authors do not show the development of hair fiber; they focused their work in the different steps of cytokinesis in the hair follicle.

SUGGESTIONS TO AUTHORS

The manuscript is well written, and it flows quite nicely. I have some comments on the text:

I did not understand why the first image cited in the work is Supp S3, and not image 1 or even S1. On image S3, I did not understand why the authors are showing 2 different cells in different magnifications. Either they show both cells in the same magnification or show a low and higher mag for both cells.

Perhaps it would be useful (and easier for readers) to get the different number of cells in the different stages of division process in table S2 with the colours that were used in figure 1H. In Fig S4, it is really hard to see in the magnification shown the different subcellular structures, more specifically nucleoli and vesicles. They appear really tiny, and one must zoom in a lot to see anything. A suggestion could be to get a supp image where the authors clearly show each structure in EM image and in 3D rendering. Are the cells represented in column Ai-Gi the same as represented in columns ii to v? If not, why not?

For the cumulative staining measurement, did the authors measure the stain intensity in each z-slice and then sum it up? Or was measured as a volume? It was not clear for this reviewer.

In Fig4, What is or what define a cluster of mitochondria? Perhaps the authors should specify what were the parameters for this. Were there any other visible organelle that the authors could measure? Perhaps cytoskeleton elements? Considering "vesicles" for the measurements, what were these? They perhaps could be anything membrane-bounded structure in the cytoplasm, being broken down Golgi, endosomes, lysosomes etc. So this category is not actually meaningful for the analysis.

The reference Hua et al (2015) cited in Methods (page 10, line 328) was not in the list of References.

Is there a reason for some of the processing steps being at high temperatures (like TCH at 40°C and lead aspartate at 50°C), while others are at very low temperature - ethanol dehydration was at 4°C. Wouldn't this thermoshock cause artifact to the samples?

Have the authors not checked the samples in a routine TEM scope prior going to the serial block-face stage? If so, this should be mentioned.

In Supp Image S2, the authors indicate they did high-pressure freezing and freeze-substitution in the samples. But this was not at all mentioned in the Methods description.

In line 362, it should be "The acquired images were aligned using the Etomo..." and not "The acquired images were aligned used the Etomo..."

Reviewer 2

SUMMARY OF THE ADVANCE MADE IN THIS PAPER AND ITS POTENTIAL SIGNIFICANCE TO THE FIELD

This is a rare detailed study in which detailed cellular processes, such as cell division and organelle activities and localisation, have been mapped using morphological methods and SBF-SEM in order to better understand dynamic physiological processes during cell division in the hair follicle. The findings provide evidence to support subcellular dynamics and close cooperation between different sub cellular structures as well as confirming directional cell division releasing daughter cells into the coordinated differentiation pathways for the hair fibre and sheaths.

Insights such as non-random positioning of chromosomes in metaphase and co-localisation with mitochondria and the role of ER/Golgi vesicles in new membrane formation add fascinating insights into cell division.

SUGGESTIONS TO AUTHORS

The authors might reflect on the possible relevance of their methods to understanding nuclear organisation and coordinated gene expression via chromatin organisation across distances during cellular processes such as cell differentiation e.g. see review by Botchkarev et al

Botchkarev VA, Fessing MY, Sharov AA. Deciphering a Message from the Nucleus: How Transcription Factors and Spatial Chromatin Interactions Orchestrate Epidermal Differentiation. *J Invest Dermatol.* 2023 Jul;143(7):1117-1120.

Major comments [Please request additional experiments only if they are essential for supporting the conclusions; authors should be encouraged to highlight any claims that are preliminary or speculative, or to discuss any pitfalls or alternative interpretations in a 'Limitations' section]

no major comments

Minor comments

Understandably there were limited samples used for this study. Fig 1H shows a graph based on n=1 follicle. Whilst representative images from n=1 follicle may be acceptable, I feel data such as in Fig 1H should be reflective of multiple follicles so we understand the variability. Similarly, it isn't clear how many cells/nuclei were used in the other figures such as Figure 2C/3C/D - can the authors state the number of nuclei in the bars of graphs in figure legend, especially where statistical comparisons are made and only n=1 is reflected?

Suppl Fig S9 correct the spelling of Pprophase

Reference to both Fig and Figure in the text. Standardise to one or the other.

Reviewer 3

SUMMARY OF THE ADVANCE MADE IN THIS PAPER AND ITS POTENTIAL SIGNIFICANCE TO THE FIELD

The manuscript presents volume electron microscopy (SBF-SEM) data of ovine hair follicles, with analysis focused on 3D reconstructing cells at various stages of cell cycle with the aim to present the first 3D ultrastructural data on cell cycle progression within tissue.

This looks to be a first example of stages of division in situ in tissue (previous published work is in monolayers of cells in culture, 2D or 3D culture), however I believe that the manuscript promises more than it delivers. Whilst I believe it would be useful for the field to see the data and analysis, I think several of the statements aren't necessarily novel, or strongly supported.

SUGGESTIONS TO AUTHORS

Major comments [Please request additional experiments only if they are essential for supporting the conclusions; authors should be encouraged to highlight any claims that are preliminary or speculative, or to discuss any pitfalls or alternative interpretations in a 'Limitations' section]

The authors present nice data on ultrastructural features of cells in various stages of division in situ in the ovine hair follicle. The manuscript is clearly written and well referenced with excellent figures, and I believe the data to be interesting and important for the community but based on my interpretation of the data as presented, too bold to be supported by the manuscript as it is written.

For example:

* The data is presented as though it is from n=2 hair follicles, but it is unclear how much of the data is actually from only 1 hair follicle, especially as one looks to be approx. 50x the size of the other dataset. This makes it hard to support statements like "Mitotically active cells were localised within the germinative matrix (Fig 1G) and exhibited directed centripetal migration as they progressed 84 through different phases (Fig 1I)." - Lines 82-84, Figure 1I, as common features in hair follicles, plural.

* The manuscript consistently refers to this data providing or aiming to investigate "dynamics" of cell division or migration or organelle movement. Eg. lines: 27, 34, 48, 66, 78, 83, 159, etc. Whilst electron microscopy can provide exceptional resolution, it is not traditionally used

for studying the dynamics of a process, as it is usually a static representation at a single time point. I believe the data shows excellent information at specific time points but does not reveal much about the dynamics of this movement, and as such the text is misleading.

* It is unclear to me, the biological interpretative value of the hotspot analysis, as this (if I understand correctly) is a 2D analysis of 3D data, that depending upon the stage of division and morphology of an individual cell could have very different interpretations.

* I am also unclear of the value of the vesicular quantification (only done in two of the three Metaphase datasets available - why?) as it is difficult for the reader to know what vesicles are being quantified (endosomes, lysosomes, Golgi, MVBs, etc) and difficult to see how these can even be seen in the data, as - from the manuscript, it is not possible to see these clearly. It is also quite hard to see the ER clearly -in the figures provided.

* I am not an expert in statistics, but I wonder if there is a potential n=1 hair follicle and n=1 cell reconstructed for that stage - what the appropriate test should be. I would be happy to see the data presented without error bars, or n increased so that appropriate statistical tests can be performed. see Fig. 4 for examples of error bars with n=1. Clarity on what the information represents is essential for reader interpretation.

Strengths:

Metaphase quantification

Karyotype reconstruction

Single cell 3D reconstructions - (without error bars)

I believe it would be useful if the manuscript could be rewritten to clearly present the data more openly, and for the authors to more clearly highlight where the data agrees and disagrees with existing datasets -and knowledge from light microscopy, and emphasize where the strengths of this technique could be for the cell cycle community.

Minor comments

Line 33 - typo details

Figure 1, panel H = cell number per what? Hair follicle? Cubic mm? non dividing cells?

Supplementary Table S1 - implies that there should have been 3 hair follicle samples included in the manuscript. This table could be improved to include x, y, z dataset dimensions in x and y (not just z).

Supplementary figure 1= is z referring to number of slices? Or um. Would be nice to understand location and proportion of a HF each dataset was able to be sampled.

First revision

Author response to reviewers' comments

Reviewer 1: SUMMARY OF THE ADVANCE MADE IN THIS PAPER AND ITS POTENTIAL SIGNIFICANCE TO THE FIELD

The manuscript by Jadav et al is a gigantic task of reconstructing hair cells in different division steps in 3D-EM. The task is not trivial and the authors did a great job putting together and analysing the data.

However, I did not see any major contribution to the field of cell biology. It is an amazing work in the VolumeEM field, but I believe the impact beyond this is very limited.

Thank you for your valuable feedback.

This study represents the first in situ 3D ultrastructural characterisation of mitosis within an intact mini-organ, rather than in 2D or 3D culture systems. By reconstructing complete karyotypes from dividing cells in this context, we provide unique insights into chromatin organization and organelle dynamics across different mitotic phases—observations that

were previously inaccessible. These cells subsequently give rise to multiple differentiated lineages, highlighting the biological relevance of our findings.

The hair follicle is an ideal model for studying stem cell and transit-amplifying cell behaviour during mammalian tissue regeneration. Our data provide a foundation for understanding epithelial self-renewal, keratinization, and hair fibre formation—questions central to developmental biology and with potential implications for biomedical and agricultural applications, including keratin-based biomaterials. Volume electron microscopy (vEM), such as Serial Block-Face SEM, enables high-resolution imaging across large tissue volumes while preserving native architecture and cell-cell interactions. This allows the capture of rare and dynamic processes, such as mitosis, in situ within the intact mini-organ.

In summary, beyond advancing vEM methods, our work provides new mechanistic insight into cell division and tissue organization in a native 3D context, bridging fundamental cell biology and applications in biomaterials and regenerative medicine. We have clarified these points in the revised manuscript and adjusted the language around generalizability to better reflect the scope and significance of our findings.

The number of samples analysed is very small, limiting any interpretation that can be made.

Yes, the samples were examined using routine TEM prior to Serial Block-Face SEM (SBF-SEM) imaging to confirm preservation of ultrastructure and the adequacy of contrast. In addition, cultured cells were processed concurrently with the hair follicles to assess and compare the quality of preservation between individual cells and those within the intact organ. This provided an additional level of validation for the sample preparation protocol.

Perhaps the title should be changed. The authors do not show the development of hair fiber; they focused their work in the different steps of cytokinesis in the hair follicle.

Thank you for pointing this out. We have changed the title to “Uncovering Mitotic Ultrastructure in the Native Hair Follicle Using Volume Electron Microscopy”

The manuscript is well written, and it flows quite nicely. I have some comments on the text: I did not understand why the cells in the different stages of division process in table S2 with the colours that were used in figure 1H.

Thank you for your valuable feedback, we have matched the colour so it is now easier to correlate figure 1H with table S2 visually.

In Fig S4, it is really hard to see in the magnification shown the different subcellular structures, more specifically nucleoli and vesicles. They appear really tiny, and one must zoom in a lot to see anything. A suggestion could be to get a supp image where the authors clearly show each structure in EM image and in 3D rendering. Are the cells represented in column Ai-Gi the same as represented in columns ii to v? If not, why not?

Thank you for your attention to this matter, we understand it is difficult observe some of the smaller structures, we wanted to concise all these into one page for a cleaner view of all the structures. We have now included an additional supplement figure to accommodate this (supplementary Fig S7) showcasing each cellular and sub-cellular component within the EM image. Regarding to Fig S4 the cells represented downwards are the same across for each line. We have now added a line to separate hoping to make this clearer for the readers and updated the figure legend for more clarity.

For the cumulative staining measurement, did the authors measure the stain intensity in each z-slice and then sum it up? Or was measured as a volume? It was not clear for this reviewer.

Thank you for your question. The stain intensity was measured by calculating the integrated density on each individual z-slice within the segmented region and then summing these values across all slices to obtain the total cumulative stain intensity for the entire volume. We have added an extra sentence to mention this in the method.

In Fig4, What is or what define a cluster of mitochondria? Perhaps the authors should specify what were the parameters for this. Were there any other visible organelle that the authors could measure? Perhaps cytoskeleton elements? Considering "vesicles" for the measurements, what were these? They perhaps could be anything membrane-bounded structure in the cytoplasm, being broken down Golgi, endosomes, lysosomes etc. So this category is not actually meaningful for the analysis.

Thank you for your comments. For each organelle (mitochondrion), the centroid was extracted and analysed using k-means clustering, which groups mitochondria based on spatial proximity. We then applied silhouette analysis and modelled the best number of clusters to determine the optimal number. These are classical widely used algorithms that allow us to identify number of clusters of organelles at different mitotic stages. It provides insightful into whether organelles are randomly distributed or form distinct number of groups across different mitotic stages.

Regarding vesicles, due to the limitations of label-free volume EM, we cannot confidently define their specific identities. Thus, we use "vesicles" as a broad morphological category. Despite their abundance and distribution across mitotic stages still provide meaningful information on cytoplasmic remodelling. This broad classification also offers a foundation for future studies incorporating molecular labelling to identify specific vesicle types. We have made sure to add this statement in the manuscript.

The reference Hua et al (2015) cited in Methods (page 10, line 328) was not in the list of References.

Thank you for pointing this out. We have rectified this mistake.

Is there a reason for some of the processing steps being at high temperatures (like TCH at 40°C and lead aspartate at 50°C), while others are at very low temperature - ethanol dehydration was at 4°C. Wouldn't this thermoshock cause artifact to the samples?

Have the authors not checked the samples in a routine TEM scope prior going to the serial block-face stage? If so, this should be mentioned.

Thank you for raising this point.

All these temperature-specific steps have been **previously established and validated** in the literature. Elevated temperatures for TCH and lead aspartate, as well as cooled dehydration, are standard practices in high-fidelity EM workflows and have been shown to preserve ultrastructure while enhancing contrast.

Elevated temperatures—specifically 40 °C for thiocarbohydrazide (TCH) and 50 °C for lead aspartate—are intentionally applied to enhance reagent penetration and optimise contrast for volume electron microscopy. Heating the TCH step improves its capacity to bridge osmium-bound tissue components, facilitating additional osmium deposition during the second osmium exposure. This results in enhanced electron density, particularly in lipid-rich structures such as mitochondrial cristae, plasma membranes, and lipid droplets. This principle underlies the OTO (osmium-TCH-osmium) method, which is known to yield sharper membrane delineation and improved resolution in vEM datasets. The historical precedent for this temperature-dependent enhancement was established by Seligman et al. (1966), who demonstrated that TCH treatment at elevated temperatures produces marked membrane contrast in lipid-rich tissues. For lead aspartate, the release of Pb²⁺ ions is accelerated at higher temperature, improving staining efficiency (Walton, 1978). Heating to 50 °C in the final stages significantly increases contrast (Polilov et al., 2021). By contrast, ethanol dehydration is performed at 4 °C to minimise lipid extraction and better preserve membrane integrity. Cooling during dehydration is a standard practice in high-fidelity EM workflows and complements the subsequent contrast-enhancing stages. Comparative studies indicate that using the ethanol series (30%, 50%, 70%, 95%, 100%) at

4 °C, followed by final dehydration in acetone or propylene oxide, provides optimal preservation (Polilov et al., 2021).

Importantly, the tissues were chemically fixed prior to the temperature-specific staining steps, which preserves ultrastructure and minimises the risk of thermal artefacts. Once fixed, the tissue is stabilised, and moderate temperature changes (within a biologically safe range) do not introduce significant ultrastructural damage.

Regarding sample quality control: Yes, the samples were examined using routine TEM prior to serial block-face SEM (SBF-SEM) imaging to confirm preservation of ultrastructure and adequacy of contrast. In addition, cultured cells were processed simultaneously along with the hair follicles to assess and compare the quality of preservation between individual cells and those within the context of the intact organ. This provided an additional level of validation for the sample preparation protocol. We agree this is an important detail and will clarify it in the Methods section.

In Supp Image S2, the authors indicate they did high-pressure freezing and freeze-substitution in the samples. But this was not at all mentioned in the Methods description.

Thank you for pointing this out. We processed samples using both conventional and high-pressure freezing (HPF) methods. As this manuscript focuses on mitosis and karyotyping, for consistency we presented data only from hair follicles processed using conventional methods, in which all mitotic phases were clearly identifiable. We have clarified this in the revised manuscript and updated the Methods section accordingly.

In line 362, it should be "The acquired images were aligned using the Etomo..." and not "The acquired images were aligned used the Etomo..."

Thank you for pointing this out. We have rectified this.

Reviewer 2: SUMMARY OF THE ADVANCE MADE IN THIS PAPER AND ITS POTENTIAL SIGNIFICANCE TO THE FIELD

This is a rare, detailed study in which detailed cellular processes, such as cell division and organelle activities and localisation, have been mapped using morphological methods and SBF-SEM in order to better understand dynamic physiological processes during cell division in the hair follicle. The findings provide evidence to support subcellular dynamics and close cooperation between different sub cellular structures as well as confirming directional cell division releasing daughter cells into the coordinated differentiation pathways for the hair fibre and sheaths.

Insights such as non-random positioning of chromosomes in metaphase and co-localisation with mitochondria and the role of ER/Golgi vesicles in new membrane formation add fascinating insights into cell division.

SUGGESTIONS TO AUTHORS

The authors might reflect on the possible relevance of their methods to understanding nuclear organisation and coordinated gene expression via chromatin organisation across distances during cellular processes such as cell differentiation e.g. see review by Botchkarev et al

Botchkarev VA, Fessing MY, Sharov AA. Deciphering a Message from the Nucleus: How Transcription Factors and Spatial Chromatin Interactions Orchestrate Epidermal Differentiation. *J Invest Dermatol.* 2023 Jul;143(7):1117-1120.

Thank you for the suggestion, but to our understanding this wouldn't be very applicable to vEM at this stage. However, we have added a sentence outlining that it would be interesting to incorporate molecular and CLEM techniques for future studies in the discussion section.

Major comments [Please request additional experiments only if they are essential for supporting the conclusions; authors should be encouraged to highlight any claims that are preliminary or speculative, or to discuss any pitfalls or alternative interpretations in a 'Limitations' section]

no major comments

Minor comments

Understandably there were limited samples used for this study. Fig 1H shows a graph based on n=1 follicle. Whilst representative images from n=1 follicle may be acceptable, i feel data such as in Fig 1H should be reflective of multiple follicles so we understand the variability. Similarly, it isn't clear how many cells/nuclei were used in the other figures such as Figure 2C/3C/D - can the authors state the number of nuclei in the bars of graphs in figure legend, especially where statistical comparisons are made and only n=1 is reflected?

Thank you for your suggestion, on the bar itself contains data points, and in supplementary Table S2 also presents the number of cells analysed in detail for both non-dividing and dividing sheep hair follicle (HF) cells. Dividing cells are further categorised by mitotic phase (prophase, metaphase, anaphase, telophase, and cytokinesis). The data were derived from three separate sample volumes taken from two individual follicles, not from a single follicle. These volumes were selected are representative sampling of the germinative matrix from both follicles used in the analysis.

Suppl Fig S9 correct the spelling of Pprophase

Thank you, we corrected this mistake

Reference to both Fig and Figure in the text. Standardise to one or the other.

Thank you, we corrected this mistake

Reviewer 3: SUMMARY OF THE ADVANCE MADE IN THIS PAPER AND ITS POTENTIAL SIGNIFICANCE TO THE FIELD

The manuscript presents volume electron microscopy (SBF-SEM) data of ovine hair follicles, with analysis focused on 3D reconstructing cells at various stages of cell cycle with the aim to present the first 3D ultrastructural data on cell cycle progression within tissue.

This looks to be a first example of stages of division in situ in tissue (previous published work is in monolayers of cells in culture, 2D or 3D culture), however I believe that the manuscript promises more than it delivers. Whilst I believe it would be useful for the field to see the data and analysis, I think several of the statements aren't necessarily novel, or strongly supported.

Thank you for your valuable feedback. We have amended our manuscripts to soften certain aspects. To our knowledge, this study represents the first in situ 3D ultrastructural characterisation of cell division stages within a complex, intact mini-organ, rather than in conventional 2D or 3D culture systems. While previous studies have imaged mitosis in cultured cells, these approaches do not capture the full spatial and cellular complexity of native tissue architecture—particularly within a self-renewing mini-organ such as the hair follicle. We strongly believe that this represents a **novel contribution**, as it enables direct observation of mitotic progression in the native environment of the germinative matrix—a spatially restricted zone that gives rise to the keratinised fibre and is central to hair formation.

Furthermore, the novelty of our dataset is strengthened by our ability to reconstruct complete karyotypes from dividing cells in situ. This level of ultrastructural detail allows us to address long-standing questions regarding nuclear architecture, chromatin dynamics, and organelle organisation during hair fibre development. These insights are also highly relevant to understanding keratin self-assembly and fibre formation, with important implications not

only for developmental biology but also for applications in the agricultural and biomedical sectors.

SUGGESTIONS TO AUTHORS

Major comments [Please request additional experiments only if they are essential for supporting the conclusions; authors should be encouraged to highlight any claims that are preliminary or speculative, or to discuss any pitfalls or alternative interpretations in a 'Limitations' section]

The authors present nice data on ultrastructural features of cells in various stages of division in situ in the ovine hair follicle. The manuscript is clearly written and well referenced with excellent figures, and I believe the data to be interesting and important for the community but based on my interpretation of the data as presented, too bold to be supported by the manuscript as it is written.

Thank you for bringing this up. It is well established that hair follicles undergo **anisotropic transformation**, in which mitotically active cells in the bulb region near the dermal papilla progressively differentiate into keratinised fibre material as they migrate upwards. This spatial organisation of proliferative and differentiating compartments has been demonstrated in numerous immunohistochemical and proteomic studies.

In our work, we **visualise these features directly at ultrastructural resolution for the first time** using volume electron microscopy (vEM), allowing us to resolve not only the spatial layout but also aspects of temporal organisation. While the current dataset is derived from two follicles—with one contributing more extensively than the other—the features observed are **consistent with well-characterised biological principles** and provide a high-resolution, structural validation of known cellular behaviours within the follicle.

For example:

* The data is presented as though it is from n=2 hair follicles, but it is unclear how much of the data is actually from only 1 hair follicle, especially as one looks to be approx. 50x the size of the other dataset. This makes it hard to support statements like "Mitotically active cells were localised within the germinative matrix (Fig 1G) and exhibited directed centripetal migration as they progressed 84 through different phases (Fig 1I)." - Lines 82-84, Figure 1I, as common features in hair follicles, plural.

The hair follicle is structurally complex and non-linear, with densely packed, concentric layers of cells from multiple lineages in the bulb region. This complexity, combined with inherent challenges in sectioning (as noted in previous reviews), means that variation can arise depending on how the tissue is cut. In our study, we captured high-resolution data on cell-cell interfaces, organelle structures, migration pathways, and tissue architecture from the base of the follicle up to the early differentiation zone for both the follicles presented. Acquiring and analysing this dataset required over a year of intensive manual and semi-automated annotation. The dataset analysed also depended on which mitotic events were captured in the given snapshot. Of all follicles we observed, anaphase—the shortest phase of mitosis—was identifiable in only one follicle, which further justifies our focus on Follicle 1 for most analyses. Observations such as the localisation of dividing cells within the germinative matrix and their apparent directed migration are consistent with known hair follicle biology.

To ensure clarity, we have revised the manuscript to indicate that these findings are representative of the follicles analysed, rather than generalised to all hair follicles. Considering the scale, internal consistency, and agreement with known biology, a sample size of one is generally acceptable in volume EM studies, and including a second follicle adds further support.

* The manuscript consistently refers to this data providing or aiming to investigate "dynamics" of cell division or migration or organelle movement. Eg. lines: 27, 34, 48, 66, 78, 83, 159, etc. Whilst electron microscopy can provide exceptional resolution, it is not traditionally used for studying the dynamics of a process, as it is usually a static representation at a single time point. I believe the

data shows excellent information at specific time points but does not reveal much about the dynamics of this movement, and as such the text is misleading.

Thank you for pointing this out. We fully acknowledge that electron microscopy captures **high-resolution static snapshots**, rather than real-time processes. However, the hair follicle is inherently a dynamic structure, undergoing continuous cycles of cell proliferation, differentiation, and degradation to produce the cornified fibre. While our vEM dataset captures a single time point, it reflects a biologically active system in the midst of these ongoing processes. Our interpretations of cell division and migration are based on the spatial distribution, morphology, and cellular context captured within this complex and temporally ordered structure. To avoid potential confusion/conflict, we have revised the manuscript to clarify that references to “dynamics” refer to biological processes inferred from spatial features

* It is unclear to me, the biological interpretative value of the hotspot analysis, as this (if I understand correctly) is a 2D analysis of 3D data, that depending upon the stage of division and morphology of an individual cell could have very different interpretations.

Thank you for your comment. We recognise that the hotspot analysis uses a 2D projection of 3D data, which may oversimplify spatial complexity and vary with cell morphology and mitotic stage. However, this approach enables standardised spatial comparisons by aligning cells along their major axis and dividing them into a consistent 3x3 matrix. While it reduces dimensionality, it effectively highlights relative spatial patterns and organelle enrichment. We note this limitation in the manuscript and view the hotspot analysis as complementary to our full 3D assessments. We also made changes in the method to clarify that it uses a 2D projected image.

* I am also unclear of the value of the vesicular quantification (only done in two of the three Metaphase datasets available - why?) as it is difficult for the reader to know what vesicles are being quantified (endosomes, lysosomes, Golgi, MVBs, etc) and difficult to see how these can even be seen in the data, as - from the manuscript, it is not possible to see these clearly. It is also quite hard to see the ER clearly -in the figures provided.

Thank you for raising up this question, unfortunately for the 3rd metaphase cell, we were unable to confidently identify small structures like the vesicles due to the hindered resolution for the particular n slice.

Regarding type of vesicles, due to the limitations of label-free volume EM, we cannot confidently define their specific identities. Thus, we use “vesicles” as a broad morphological category. We have made sure to add this statement in the manuscript in the method. Additionally, we have now included an additional supplement figure to accommodate this (supplementary F S7) showcasing each cellular and sub-cellular component zoomed up within the EM image.

* I am not an expert in statistics, but I wonder if there is a potential n=1 hair follicle and n=1 cell reconstructed for that stage - what the appropriate test should be. I would be happy to see the data presented without error bars, or n increased so that appropriate statistical tests can be performed. see Fig. 4 for examples of error bars with n=1. Clarity on what the information represents is essential for reader interpretation.

Thank you for raising this important point. In Figure 4, error bars are only included when more than three cells were analysed for a given measurement. For cases where $n = 1$, we show only a single dot, which represents the measurement from one cell. These points were included to provide transparency and illustrate the spread of the available data. We agree that increasing n would yield more robust statistical conclusions; however, due to the time-intensive nature of segmentation and reconstruction, we were limited in the number of cells that could be fully analysed. To partially address this limitation and improve statistical power, we also compared dividing cells (all mitotic phases pooled) with non-dividing cells, thereby increasing n and allowing more meaningful statistical testing.

Strengths:

Metaphase quantification

Karyotype reconstruction

Single cell 3D reconstructions - (without error bars)

I believe it would be useful if the manuscript could be rewritten to clearly present the data more openly, and for the authors to more clearly highlight where the data agrees and disagrees with existing datasets -and knowledge from light microscopy, and emphasize where the strengths of this technique could be for the cell cycle community.

Great suggestion, we have added made changes and added to our discussion.

Minor comments

Line 33 - typo details

Thank you, resolved.

Figure 1, panel H = cell number per what? Hair follicle? Cubic mm? non dividing cells?

Thank you, we have modified the title as “frequency of mitotically dividing cells across the hair follicle” and y-axis title to “cell count”.

Supplementary Table S1 - implies that there should have been 3 hair follicle samples included in the manuscript. This table could be improved to include x, y, z dataset dimensions

Thank you for picking up this mistake! We have corrected this and included the XY-dimension.

Supplementary figure 1= is z referring to number of slices? Or um. Would be nice to understand location and proportion of a HF each dataset was able to be sampled.

Thank you, it is refereeing to the number slices from the z-stack. We have added this to the figure legend.

References

Mikula S, Denk W. High-resolution whole-brain staining for electron microscopic circuit reconstruction. *Nat Methods*. 2015 Jun;12(6):541-6. doi: 10.1038/nmeth.3361. Epub 2015 Apr 13. PMID: 25867849.

Deerinck TJ, Shone TM, Bushong EA, Ramachandra R, Peltier ST, Ellisman MH. High-performance serial block-face SEM of nonconductive biological samples enabled by focal gas injection-based charge compensation. *J Microsc*. 2018 May;270(2):142-149. doi: 10.1111/jmi.12667. Epub 2017 Dec 1. PMID: 29194648; PMCID: PMC5910240.

Seligman AM, Wasserkrug HL, Hanker JS. A new staining method (OTO) for enhancing contrast of lipid-containing membranes and droplets in osmium tetroxide-fixed tissue with osmiophilic thiocarbonylhydrazide(TCH). *J Cell Biol*. 1966 Aug;30(2):424-32. doi: 10.1083/jcb.30.2.424. PMID: 4165523; PMCID: PMC2106998.

Polilov, A.A., Makarova, A.A., Pang, S. *et al*. Protocol for preparation of heterogeneous biological samples for 3D electron microscopy: a case study for insects. *Sci Rep* 11, 4717 (2021). <https://doi.org/10.1038/s41598-021-83936-0>

Second decision letter

MS ID#: jcs.264198R1

MS TITLE: Uncovering Mitotic Ultrastructure in the Native Hair Follicle Using Volume Electron Microscopy

AUTHORS: Mihnea Bostina; Nickhil Jadav; Sailakshmi Velamoor; Niki Hazelton; Karen Reader; Duane Harland

ARTICLE TYPE: Research Article

Dear Dr Bostina,

We have now reached a decision on the above manuscript.

As one of the original reviewers was unavailable we have received just two reviews. One reviewer was happy with the revised manuscript but the other felt the manuscript was unsuitable for a general cell biology journal. They also made some suggestions to improve the manuscript. I appreciate the concerns of this reviewer but I am willing to consider a further revision which addresses these issues. It will also be important to consider their concerns about the importance of the work to the general cell biology readership of the Journal of Cell Science.

Reviewer 1

SUMMARY OF THE ADVANCE MADE IN THIS PAPER AND ITS POTENTIAL SIGNIFICANCE TO THE FIELD

This is a much better version of the manuscript; and I thank the authors for bringing such work together and for considering the suggestions made by the reviewers.

I still think the overall outcomes from this study (and there are several and high quality data/image analysis) are interesting to a limited biological field; and perhaps it would be better fitted to a more technological journal. But I can see the points raised by the authors and the other reviewers.

The authors cited some references in their reply to the reviewers - perhaps they should consider adding those in the main text of the manuscript.

The authors mentioned that they had added the information about thin sectioning in the Methods but I couldn't see it. If they used a transmission electron microscope to observe thin sections of the material prior to the SBF-SEM step, they must add this information.

I would suggest the authors to also add in supplemental material the scripts used for the analysis. Or even better a github repository of the scripts. This would be fantastic for other microscopists/image analysts out there.

Reviewer 2

SUMMARY OF THE ADVANCE MADE IN THIS PAPER AND ITS POTENTIAL SIGNIFICANCE TO THE FIELD

This is a highly detailed study of mitosis in the ovine hair follicle using novel methods. Hair follicle matrix cells are amongst the most rapidly dividing cells in the body. Studying the dynamics of mitosis is very difficult yet this paper describes not only mitosis but the association of other organelles with the process, which will be of interest to hair biologists. Although the number of follicles studied is limited, this is not unusual where electron microscopy is involved and justified during review.

SUGGESTIONS TO AUTHORS

Major comments [Please request additional experiments only if they are essential for supporting the conclusions; authors should be encouraged to highlight any claims that are preliminary or speculative, or to discuss any pitfalls or alternative interpretations in a 'Limitations' section]

I have no further major comments.

The authors addressed in details the comments from reviewers and made appropriate adjustments to the manuscript including limitations of the methods and interpretation

Minor comments

no more comments here

Second revision

Author response to reviewers' comments

Reviewer 1: SUMMARY OF THE ADVANCE MADE IN THIS PAPER AND ITS POTENTIAL SIGNIFICANCE TO THE FIELD

This is a much better version of the manuscript; and I thank the authors for bringing such work together and for considering the suggestions made by the reviewers. I still think the overall outcomes from this study (and there are several and high-quality data/image analysis) are interested to a limited biological field; and perhaps it would be better fitted to a more technological journal. But I can see the points raised by the authors the other reviewers. The authors cited some references in their reply to the reviewers - perhaps they should consider adding those in the main text of the manuscript.

Thank you for your positive assessment and constructive suggestions. We have now incorporated the references in the response into the main manuscript text, and the following modifications have been highlighted.

The authors mentioned that they had added the information about thin sectioning in the Methods but I couldn't see. If they used a transmission electron microscope to observe thin sections of the material prior the SBF-SEM step, they must add this information. I would suggest the authors to also add in supplemented material the scripts used for the analysis. Or even better a github repository of the scripts. This would be fantastic for other microscopists/image analysers out there.

Thank you for pointing this out. The thin-sectioning and TEM imaging procedures have now been added to the Methods section, including details of sample preparation, section thickness, staining, and the transmission electron microscope and imaging parameters used (Line 118 to 125; Supplementary Figure S11).

The analysis scripts used in this study are part of a separate manuscript that is currently in preparation/submitted and under review, where they are described and documented in detail. For this reason, we are unable to make them publicly available at this stage. However, we have set up a GitHub repository, so the readers will be able to follow the link from the manuscript (Line: 207-209). Thank you again for this suggestion.

Reviewer 2: SUMMARY OF THE ADVANCE MADE IN THIS PAPER AND ITS POTENTIAL SIGNIFICANCE TO THE FIELD

This is a highly detailed study of mitosis in the ovine hair follicle using novel methods. Hair follicle matrix cells are amongst the most rapidly dividing cells in the body. Studying the dynamics of

mitosis is very difficult yet this paper describes not only mitosis but the association of other organelles with the process, which will be of interest to hair biologists. Although the number of follicles studies is limited, this is not unusual where electron microscopy is involved and justified during review.

Thank you for the review and supportive feedback.

Third decision letter

MS ID#: jcs.264198R2

MS Title: Uncovering Mitotic Ultrastructure in the Native Hair Follicle Using Volume Electron Microscopy

Authors: Mihnea Bostina; Nickhil Jadav; Sailakshmi Velamoor; Niki Hazelton; Karen Reader; Duane Harland

Article Type: Research Article

Dear Dr Bostina,

I am happy to tell you that your manuscript has been accepted for publication in Journal of Cell Science, pending standard publication integrity checks.